FastST: an efficient tool for inferring decomposition and directionality of microbial communities

Choi Joung Min 1
Wu Xiaowei 2
Zhang Liqing lqzhang@cs.vt.edu 1
1 Department of Computer Science, Virginia Polytechnic Institute and State University (Virginia Tech) , Blacksburg , VA , United States of America
2 Department of Statistics, Virginia Polytechnic Institute and State University (Virginia Tech) , Blacksburg , VA , United States of America
Cheung Simon Man Kit
Electronic publication date: 2025 Oct 27
Publication date: 2025
Volume: 13
Electronic Location ID: e20161
Received 2025 May 13; Accepted 2025 Sep 10
Copyright: ©2025 Choi et al.
Copyright year: 2025
Copyright holder: Choi et al.
License: This is an open access article distributed under the terms of the Creative Commons Attribution License, which permits unrestricted use, distribution, reproduction and adaptation in any medium and for any purpose provided that it is properly attributed. For attribution, the original author(s), title, publication source (PeerJ) and either DOI or URL of the article must be cited.
License URL: https://creativecommons.org/licenses/by/4.0/

Keywords: Microbial source tracking, Microbiome, Generalized least squares

Funding: National Science Foundation (NSF) #2004751 #2125798 #2344169 #2319522 The National Institutes of Health (NIH) #1R01AI179686-01A1 This work was funded by National Science Foundation (NSF) #2004751, #2125798, #2344169 and #2319522, as well as the National Institutes of Health (NIH) grant #1R01AI179686-01A1. There was no additional external funding received for this study. The funders had no role in study design, data collection and analysis, decision to publish, or preparation of the manuscript.

==============================
Microbiomes play crucial roles in human health, disease development, and global ecosystem functioning. Understanding the origins, movements, and compositions of microbial communities is essential for unraveling the principles governing microbial ecology. Microbial source tracking (MST) approaches have emerged as valuable tools for quantifying the proportions of different microbial sources within target communities, enabling researchers to track transmissions between hosts and environments, identify similarities between microbiome samples, and determine sources of contamination in various settings. Current MST methods like SourceTracker2 and FEAST have advanced the field by employing Bayesian and expectation-maximization approaches, respectively, but are limited by computational inefficiency with high-dimensional data and inability to infer directionality in source-sink relationships. This study presents a novel computational framework for microbial source tracking called FastST. FastST infers the relative contributions of source environments to sink microbiomes while also determining directionality when source-sink relationships are not predefined. Through extensive simulation studies with varying numbers of sources and complexity, FastST demonstrates superior performance in both accuracy and computational efficiency compared to FEAST and SourceTracker2, maintaining consistent execution times even as the number of source environments increases. Furthermore, the proposed method achieved over 90% accuracy in directionality inference across all tested scenarios, even when multiple major sources are present, broadening its applicability in practical microbiome research and environmental monitoring. FastST and data simulation codes are publicly available at https://github.com/joungmin-choi/FastST.

Introduction

The microbiome refers to a collection of microorganisms inhabiting a specific environment, which forms complex and diverse communities of multiple interacting species (Castro-Nallar et al., 2015). Previous studies have demonstrated the profound impact of the microbiome in complex diseases such as diabetes, inflammatory bowel disease, colorectal cancer, and allergy outcomes (Marchesi et al., 2011; Qin et al., 2012; Kostic, Xavier & Gevers, 2014; Fujimura & Lynch, 2015), which indicates the potential of human microbes as biomarkers for disease diagnosis or as therapeutic targets for treatment (Manor et al., 2020). Microbes also play a crucial role in various environments, along with the capacity to maintain a healthy global ecosystem responding to the climate change (Cavicchioli et al., 2019) and participating in essential biogeochemical cycling events, such as carbon and nitrogen fixation (Gougoulias, Clark & Shaw, 2014).

Prediction of the structure and dynamics of the microbial community is essential for understanding microbiome development and ecosystem function (Meroz et al., 2021). Insights into the origins of microbial communities and the movement of microbes across different ecosystems can help unravel the rules that govern microbial ecology. The key, albeit challenging, step in this analysis is to characterize the composition of microbial communities, as they typically comprise multiple source environments, including various contaminants and other microbial communities that have interacted with the sampled habitat (Shenhav et al., 2019). To resolve this, microbial source tracking (MST) approaches have been presented, which aim to quantify the proportion of different microbial samples (sources) in a target microbial community (sink) (Shenhav et al., 2019; Knights et al., 2011; McGhee et al., 2020). Through microbial source tracking (MST), transmissions between different hosts and environments can be tracked and similarities between microbiome samples can be identified (Briscoe, Halperin & Garud, 2023). MST can be applied to the determination of discrete sources of fecal pollution (Unno et al., 2018), tracking bacterial contamination routes of municipal water (Liu et al., 2018) and natural fresh water systems including rivers and lakes (Staley et al., 2018), and disease prevention (Fu & Li, 2014).

The most recent approaches for microbial source tracking includes SourceTracker2 (Knights et al., 2011), FEAST (Shenhav et al., 2019), STENSL (An et al., 2022), and SourceID-NMF (Huang, Cai & Sun, 2024) methods. These methods use species abundance profiles of the sample of interest and potential sources to calculate the percentages of sinks that are attributable to each potential source. Based on a Bayesian framework, SourceTracker2 employs Markov chain Monte Carlo (MCMC) to estimate contamination proportions in metagenomics studies. On the other hand, FEAST determines the fraction of each source environment in a target microbial community by modeling mixture proportions for various source microbial samples in a given sink sample, using expectation maximization for improved computational efficiency. STENSL, derived from the FEAST algorithm, incorporates unsupervised source selection via least-squares optimization with L1-norm regularization. SourceID-NMF employs non-negative matrix factorization (NMF) to identify source contributions to sink communities.

Despite their advancements, current microbial source tracking approaches have limitations that warrant attention for practical applications. Both SourceTracker2 and FEAST are computationally expensive, especially when handling high-dimensional microbiome data, though the latter is considered relatively efficient. Furthermore, these approaches lack the capability to infer directionality, i.e., the requirement to predetermine which is the source/sink prior to analysis. In certain real-world microbiome studies—for example, in hospital sink drains (Laço et al., 2025) and newly opened hospital wards (Lax et al., 2017)—it can be unclear which community is acting as the source and which as the sink, underscoring the critical need for methods to discern the directionality of microbial exchange. Addressing these limitations will enhance the reliability and applicability of microbial source tracking methods.

In the present study, we propose a novel inference framework, named FastST. FastST is able to accurately and efficiently estimate the relative contributions of the sources to the sink of interest, as well as infer the directionality if source and sink relationships are not known a priori. The estimation accuracy and efficiency are achieved by leveraging the generalized least squares (GLS) inference based on a multiple linear regression model. The inference of directionality is further made by selecting the maximized joint likelihood for observing the sink and source data across all enumerated Bayes network models. Through simulations, we demonstrate the outperformance of FastST in terms of mean absolute error and running time, as compared to SourceTracker2, FEAST, STENSL, and SourceID-NMF. In addition, simulations also show that FastST is able to identify the directionality in the source and sink environments.

Methods

Consider a microbiome data of K + 1 observed samples, each consisting of N taxa. We assume that, among the K + 1 samples, one is treated as the sink and the rest as K sources. Denote the observed sink and source data by x = (x1, …, xN)T and yi = (yi1, …, yiN)T, 1 ≤ i ≤ K, respectively, based on the corresponding random variable model: X∼multinomC,β1,…,βNT

Yi∼multinomCi,γi1,…,γiNT,1≤i≤K

where C=∑j=1Nxj,Ci=∑j=1Nyij are the total taxa counts of the sink and the ith source, and βj, γij denote the relative abundance of taxa j in the sink and the ith source, respectively. Further assume that, the sink-source relationships are determined by the following mixed-proportion model: (1) βj= ∑i=0Kαiγij,1≤j≤N.

In this model, an unobserved source Y0 ∼ multinom(C0, (γ01, …, γ0N)T) is assumed to contribute to the sink with a proportion α0. Our goal is to unveil the dependence structure between the sink and the sources by estimating the parameters αi based on the observed data x and yi, for 1 ≤ i ≤ K.

Generally speaking, the estimation of αi can be achieved by maximizing the joint likelihood of observing (x, y1, …, yK). Since the mixed-proportion model depends on several latent variables, i.e., the taxa abundances βj, γij in the sink and sources and the variables in the unobserved source, the estimation is usually carried out iteratively. Typical examples include finding (local) maximum likelihood estimates (MLE) through the Expectation–Maximization (EM) algorithm (Shenhav et al., 2019), or finding maximum a posteriori (MAP) estimates by Gibbs sampling (Knights et al., 2011). Although convergence is generally ensured for the MLE or maximum a posteriori (MAP) estimation, from a practical perspective, an obvious disadvantage of these iterative methods lies in their lack of computational efficiency, especially when the number of latent or unobserved variables is high. This greatly affects the practical use of source tracking in high-dimensional microbiome data. To address this issue, we simplify the mixed-proportion model by assuming γij to be known (if not, they can be approximated by the sample estimates yi/Ci), for 1 ≤ i ≤ K, 1 ≤ j ≤ N, and propose to estimate αi’s by the generalized least square (GLS) method. This approach is described below.

Multiplying C to both side of Eq. (1), we have EXj= ∑i=0KCαiγij,1≤j≤N.

For the unobserved source Y0, since both its taxa abundance (γ01, …, γ0N)T and its proportion α0 are unknown, the estimation of these parameters certainly encounters a nonidentifiability issue. In addition, it is reasonable to treat Y0 as a nuisance contributor to the sink with a negligible proportion, so we may assume that the product α0γ0j is common across 1 ≤ j ≤ N. Letting α ~0=Cα0γ0j,1≤j≤N and α ~i=Cαi,1≤i≤K, the above equation describes a multiple linear regression model (2) Xj=α ~0+ ∑i=1Kα ~iγij+ϵj,1≤j≤N,

where ϵj denote the random error term with E[ϵj] = 0 and Covϵ=Cβ11−β1−Cβ1β2⋯−Cβ1βN−Cβ1β2Cβ21−β2⋯−Cβ2βN⋮⋮⋱⋮−Cβ1βN−Cβ2βN⋯CβN1−βN.

Replacing βj by its sample estimate xj/C and denoting this approximated Cov(ϵ) by Σ, we obtain the GLS estimate of α ~=α ~0,α ~1,…,α ~KT by α ~ ˆ=γTΣ−1γ−1γTΣ−1x,

where γ is an N × (K + 1) design matrix with first column being a vector of 1’s. Consequently, the estimate of α = (α0, α1, …, αK)T is (3) α ˆ=1CγTΣ−1γ−1γTΣ−1x.

It is known that, as a multivariate covariance matrix, Cov(ϵ) is positive semidefinite and does not have a unique inverse. To overcome this difficulty and calculate the GLS estimate of α ~, we may either replace Cov(ϵ) with a non-singular matrix by removing any row and corresponding column (and accordingly reduce dimension for γ and x) (May & Johnson, 1998), or alternatively use the Moore–Penrose inverse (pseudoinverse) of Cov(ϵ) (Tanabe & Sagae, 1992).

In practice, it is often not clear which of the K + 1 samples in the microbiome data plays the role of the sink. Denoting the samples by {S1, …, SK+1}, we define the kth model, 1 ≤ k ≤ K + 1, by

Modelk:X=Sk,yi∈Sk−,1≤i≤K,

where Sk−: = {S1, …, SK+1}∖Sk. Our purpose is to determine the correct model from a total of K + 1 candidate models so that the true source and sink relationships are testified. For convenience, let us represent the directionality of Model k by Sk− → Sk. Using a Bayesian network setting, we write the joint likelihood of observing (x, y1, …, yK) in Model k as (4) PrX,y1,…,yK=PrX|y1,…,yKPry1,…,yK=PrX|y1,…,yK∏i=1kPryi,

where the conditional likelihood Pr(x|y1, …, yK) is calculated by using the estimated parameters from Eq. (3). We then choose the correct model by maximizing Pr(x, y1, …, yK).

We note that, this directionality inference method employs the concept of structure learning for Bayesian networks. Briefly speaking, a Bayesian network is a graphical model that represents a set of random variables and their conditional dependencies via a directed acyclic graph (DAG), and structure learning for Bayesian networks refers to learning the structure of the DAG from data. As shown in Fig. 1, we model the microbiome data by a simple Bayesian network which connects each source yi, 1 ≤ i ≤ K to the sink x with a directed edge. Our directionality inference is essentially a score-based approach which uses the log-likelihood of the data under the graph structure as a criterion (or score function) to evaluate how well the Bayesian network fits the data. In practice, score-based approaches usually introduce a regularization term (e.g., Akaike Information Criterion (AIC), Bayesian Information Criterion (BIC)) to penalize overfitting of the model and favor simpler models. However, for our directionality inference problem, the regularization term can be omitted safely in the score function because all candidate graphical models are of the same complexity—it is just a matter which variable plays the role of the sink.

Figure 1 A schematic plot of the graphical model used for directionality inference.

Results

Experimental design for performance evaluation

To evaluate the proposed method, we designed three experimental settings encompassing both simulated and real microbiome data. For simulations in Scenarios 1(a) and 2, we assume N > K, which means that the number of taxa is always greater than the number of source samples. The case of N ≤ K will be discussed separately in “Discussion and conclusions”. The three evaluation scenarios are as follows:

Scenario 1: Evaluation using fully simulated microbiome data. This includes two sub-cases: (a) The sink and sources are clearly defined; that is, the directionality of the microbiome community is known; (b) The source and sink relationships are not known a priori and need to be inferred.

Scenario 2: Evaluation using a semi-synthetic dataset generated based on real microbial taxa distributions.

Scenario 3: Evaluation using real microbiome data.

In Scenario 1, the number of sources K varies from 2 to 100, and we assume that there are several major sources that contribute to the sink, each major source accounting for at least 10% of the proportions and their summed proportions fixed at 90%. The true relative abundance parameters γij and proportion parameters αi, 0 ≤ i ≤ K, 1 ≤ j ≤ N, are drawn from prespecified Dirichlet distributions. The microbiome data were then generated from the multinomial distributions and the aforementioned mixed-proportion model (Eq. (1)).

In Scenario 2, we adopt a similar generative approach as in Scenario 1, with K varying from 2 to 50 and major sources collectively accounting for 90% of the total proportion. However, the microbial taxa distribution is derived from a real microbiome dataset published by Knights et al. (2011), which includes 180 barcoded pyrosequencing samples of bacterial 16S rRNA gene sequences collected from diverse environments such as human skin, oral cavities, feces, and temperate soils (referred as “Knights et al. dataset”).

In Scenario 3, we directly evaluate our method using Knights et al. dataset, which also provided additional four 16S rRNA microbiome samples labeled as sinks—representing surface contamination from two research laboratories, a hospital, and an office building. These samples are commonly used to test MST tools. While the true source proportions are unknown in this real-world setting, the experiment enables comparison of the outputs generated by different MST methods.

Scenario 1: evaluation on fully simulated microbiome data

Source proportion estimation with known directionality

Using the simulated data in Scenario 1, we obtained an estimate of the source proportions αi, 1 ≤ i ≤ K, and calculated the mean absolute error (MAE), Jensen–Shannon divergence (JSD) and Pearson correlation (PCC) of the estimated source proportions. Repeating the simulation 1000 times, we report the average MAE for estimating the proportions of the observed sources and the major sources, as well as the contribution of the unknown source. We also reported the average JSD and PCC of the observed sources. For comparison, the average performance of the other four source tracking methods—FEAST, SourceTracker2, STENSL, and SourceID-NMF—are also included.

As summarized in Table 1 and Supplementary Material S1, FastST consistently outperformed the other two microbial source tracking methods in estimating proportions of both major and unobserved sources. FastST achieved the MAE of 0.0164 for major sources and 0.0468 for unobserved sources in simulations involving two observed sources (K = 2). FEAST showed the second-best performance, with average MAEs of 0.0474 for major sources and 0.0895 for unobserved sources, followed by SourceTracker2, STENSL, and SourceID-NMF.

Table 1 Average MAE for estimating the proportion of observed sources, the proportion of major sources, and the contribution of unobserved sources in fully simulated microbiome data across source-tracking methods.

Number of
observed sources	Number of
major sources	Sources	FastST	FEAST	SourceTracker2	STENSL	SourceID-NMF	
		Observed	0.0164	0.0474	0.1458	0.0658	0.0919	
2	2	Major	0.0164	0.0474	0.1458	0.0658	0.0919	
		Unobserved	0.0468	0.0895	0.2890	0.1305	0.1085	
		Observed	0.0033	0.0230	0.0932	0.0318	0.0870	
5	2	Major	0.0047	0.0461	0.1728	0.0618	0.1694	
		Unobserved	0.0050	0.0934	0.2262	0.1565	0.2402	
		Observed	0.0060	0.0125	0.0398	0.0333	0.0219	
5	5	Major	0.0060	0.0125	0.0398	0.0333	0.0219	
		Unobserved	0.0121	0.0374	0.1710	0.1653	0.0267	
		Observed	0.0023	0.0110	0.0696	0.0110	0.0536	
10	2	Major	0.0035	0.0363	0.2013	0.0389	0.2048	
		Unobserved	0.0070	0.0765	0.1092	0.0890	0.2849	
		Observed	0.0027	0.0118	0.0525	0.0160	0.0315	
10	5	Major	0.0030	0.0163	0.0639	0.0268	0.0447	
		Unobserved	0.0022	0.0644	0.1135	0.1500	0.1314	
		Observed	0.0017	0.0023	0.0292	0.0032	0.0074	
50	2	Major	0.0032	0.0155	0.3688	0.0279	0.0892	
		Unobserved	0.0042	0.0298	0.0168	0.0481	0.0598	
		Observed	0.0020	0.0026	0.0294	0.0032	0.0063	
50	5	Major	0.0029	0.0085	0.1490	0.0121	0.0297	
		Unobserved	0.0025	0.0289	0.0180	0.0489	0.0285	
		Observed	0.0016	0.0013	0.0169	0.0195	0.0034	
100	2	Major	0.0031	0.0115	0.4239	0.4446	0.0583	
		Unobserved	0.0020	0.0126	0.0089	0.0011	0.0052	
		Observed	0.0018	0.0015	0.0165	0.0116	0.0031	
100	5	Major	0.0030	0.0071	0.1661	0.1080	0.0216	
		Unobserved	0.0019	0.0126	0.0091	0.0960	0.0040	

Notably, when simulations were expanded to five sources (K = 5) with two major sources, FastST further improved performance, exhibiting significantly lower average MAEs of 0.0047, 0.0033, and 0.0050 for major, observed, and unobserved source proportions, respectively. Conversely, FEAST’s performance either remained similar or deteriorated compared to the two-source simulation, showing higher average MAEs of 0.0461, 0.0230, and 0.0934 for major, observed, and unobserved sources, respectively.

When the number of observed sources increased beyond ten (K ≥ 10), FEAST’s performance in estimating observed source proportions became more comparable to FastST’s. However, considerable differences persisted in estimating major and observed sources. For instance, in a simulation dataset with 50 observed sources, including two major sources, FastST yielded average MAEs of 0.0017 for observed sources, whereas FEAST had 0.0023. More significantly, for major and unobserved sources, FastST’s average MAEs were 0.0032 and 0.0042, respectively, while FEAST demonstrated substantially higher MAEs of 0.0155 and 0.0298.

Inference of directionality

Next, we evaluated FastST’s ability to infer directionality when source–sink relationships are unknown (Scenario 1(b)). Treating each of the K + 1 samples as the sink, we obtained K + 1 candidate models and calculated the joint likelihood of each model according to Eq. (4). The directionality from the sources to the sink was then inferred by selecting the model with the maximum likelihood. Repeating this simulation 100 times, we report the accuracy rate of directionality inference in Table 2. We see that, FastST is able to determine the true sink with high accuracy rate (nearly 100%) in all settings. In particular, when there are five major sources in microbiome data, directionality inference becomes more accurate because the conditional likelihood Pr(x|y1, …, yK) in this case tends to play a dominant role in the joint likelihood Pr(x, y1, …, yK). We note that, the directionality inference is a general method, not tool specific. To assess generalizability, the same simulation procedure was applied to other MST-based approaches, and we found that it similarly enabled accurate directionality inference across methods.

Table 2 Accuracy rate for inferring the sink correctly in 100 simulations using FastST, FEAST and SourceTracker2.

Number of
observed sources	Number of
major sources	FastST	FEAST	SourceTracker2	
2	2	100%	100%	100%	
5	2	100%	100%	100%	
5	5	100%	100%	100%	
10	2	100%	100%	100%	
10	5	100%	100%	100%	
50	2	100%	100%	100%	
50	5	100%	100%	100%	
100	2	99%	100%	100%	
100	5	100%	100%	100%	

Computation time comparison

We compared the computational efficiency of FastST with four other microbial source tracking methods by measuring the run-time required to complete 1,000 simulation experiments in Scenario 1(a). Tests were conducted on a single-node server equipped with an Intel Core i5 CPU (two GHz, two cores) and 16 GB of RAM. Table 3 illustrates that FastST exhibited the shortest run-time, completing simulations in 439.21 s for 10 known sources and 464.17 s for 50 known sources, each scenario containing two major sources. FEAST demonstrated the second-best performance for for smaller numbers of known sources (K ≤ 10), completing simulations in 6,543.67 s with 10 known sources, but experienced a significant increase in computational time to 61,463.39 s with 50 known sources. STENSL exhibited the longest run-times in most scenarios, recording 103,425.12 s and 42,998.03 s for 10 and 50 known sources, respectively. The results also highlight a substantial increase in computation time for FEAST and SourceID-NMF as the number of known sources increased from 10 to 50, whereas FastST maintained relatively stable performance with minimal increase. Interestingly, as the number of known sources exceeded 10, the run-time gap between FEAST and SourceTracker2 decreased. Specifically, SourceTracker2 showed consistently stable performance across all tested values of K, and significantly outperformed FEAST in scenarios with a large number of known sources (K ≥ 50), demonstrating considerably lower computation times under these conditions.

Table 3 Computation times (sec) for completing 1,000 simulations using different source tracking methods.

Number of
observed sources	Number
of major sources	FastST	FEAST	SourceTracker2	STENSL	SourceID-NMF	
2	2	414.28	2,117.23	26,320.47	1,618,385.96	429,917.70	
5	2	436.36	3,353.01	25,984.78	2,490,530.77	474,461.68	
5	5	447.26	3,107.59	25,965.82	2,702,331.68	409,722.50	
10	2	439.21	6,543.67	25,845.14	103,425.12	516,393.17	
10	5	450.41	8,015.65	25,772.28	107,073.26	435,312.15	
50	2	464.17	61,463.39	26,046.73	42,998.03	995,828.46	
50	5	479.18	79,108.87	25,977.48	62,145.34	844,902.18	
100	2	494.76	66,407.89	26,757.72	3,014,520.28	1,392,889.40	
100	5	513.71	91,033.33	26,590.93	3,532,209.32	1,403,480.23	

Scenario 2: evaluation on semi-synthetic microbiome data

To further test generalizability, we evaluated FastST using semi-synthetic datasets generated from real microbial taxonomic profiles. These datasets preserve realistic abundance patterns while allowing controlled variation in source compositions. We used the taxonomic distributions from the Knights et al. dataset to generate source and sink samples. As in Scenario 1, mixtures were simulated using Dirichlet-multinomial models with K sources and predefined proportions.

Over 100 simulations, we evaluated FastST’s source proportion estimation performance using the same metrics as in Scenario 1. The results (Table 4, Supplementary Material S2) confirmed that FastST remains similar performance to other MST tools, even when trained on real-world microbial community structures. In terms of estimating the proportion of unknown sources, FastST shows much improvement compared to the other comparison tools.

Table 4 Average MAE for estimating the proportion of observed sources, the proportion of major sources, and the contribution of unobserved sources in semi-synthetic microbiome data across source-tracking methods.

Number of
observed sources	Number of major sources	Sources	FastST	FEAST	SourceTracker2	STENSL	SourceID-NMF	
		Observed	0.0767	0.0634	0.0397	0.3275	0.2184	
2	2	Major	0.0767	0.0634	0.0397	0.3275	0.2184	
		Unobserved	0.0001	0.0795	0.0794	0.6550	0.4368	
		Observed	0.0449	0.0600	0.0099	0.1665	0.1074	
5	2	Major	0.0852	0.1283	0.0161	0.3791	0.2314	
		Unobserved	1.0 × 10−5	0.2161	0.0176	0.8327	0.5371	
		Observed	0.0507	0.0343	0.0181	0.1726	0.0981	
5	5	Major	0.0507	0.0343	0.0181	0.1726	0.0981	
		Unobserved	4.1 × 10−5	0.1383	0.0733	0.8629	0.4907	
		Observed	0.0301	0.0394	0.0098	0.0855	0.0560	
10	2	Major	0.0904	0.1390	0.0235	0.3837	0.2360	
		Unobserved	3.9 × 10−6	0.1749	0.0064	0.8551	0.5596	
		Observed	0.0342	0.0450	0.0214	0.0955	0.0559	
10	5	Major	0.0539	0.0697	0.0295	0.1740	0.0952	
		Unobserved	4.7 × 10−6	0.1505	0.0783	0.9550	0.5592	
		Observed	0.0202	0.0122	0.0070	0.0186	0.0043	
50	2	Major	0.1149	0.1517	0.0829	0.4164	0.0444	
		Unobserved	3.7 × 10−7	0.0643	0.0024	0.9310	0.0511	
		Observed	0.0189	0.0110	0.0100	0.0197	0.0081	
50	5	Major	0.0703	0.0538	0.0493	0.1776	0.0351	
		Unobserved	4.3 × 10−7	0.0537	0.0047	0.9859	0.0208	

Scenario 3: evaluation on real microbiome data

Finally, we evaluated FastST in a fully real-world setting using the Knights et al. dataset, which comprises four 16S rRNA microbiome samples from environmental surfaces—one from a hospital, two from research laboratories, and one from an office building—designated as sinks, along with 180 source samples collected from diverse environments, including human skin, oral cavities, feces, and temperate soils. In this case, the ground-truth source proportions were unknown; therefore, we focused on comparing the patterns of estimated contributions across different tools.

From the result (Fig. 2) for most sink samples, FastST produced source contribution estimates largely consistent with those from FEAST, with only minor discrepancies. Both methods indicated substantial contributions from skin and soil sources for the laboratory and office sinks, while the NICU sample exhibited a broader diversity of contributors. In contrast, FEAST and SourceTracker2 often assigned a dominant proportion of the composition to a single environment type, most frequently skin. STENSL generated patterns similar to FEAST, reflecting its algorithmic derivation from FEAST. Meanwhile, SourceID-NMF showed limited diversity in inferred sources and failed to generate results for two sink samples (Lab 1 and Lab 2), leading to the entire proportion being attributed to unknown sources.

Figure 2 Estimated source contributions to sink samples using different MST methods on the Knights et al. dataset.

Estimated contributions of the unobserved source were not reported.

Discussion and Conclusions

In this study, we introduced FastST, a novel microbial source tracking method designed to address the limitations associated with existing approaches such as FEAST, SourceTracker2, STENSL, and SourceID-NMF. FastST predicts the contributions of the sources to the sink, and infers the directionality if the source and sink relationships are not pre-defined. One unique feature of FastST is that it transforms the mixed-proportional sink-source relationships into a multiple linear regression model, which greatly simplifies the inference procedure. The source contributions were predicted by the standard GLS method and the directionality was inferred by selecting the maximized joint likelihood of a Bayes network model. Through simulations with different number of known and major sources, we demonstrated that FastST was able to achieve significant improvements in computational efficiency and accuracy, particularly evident in scenarios involving a large number of known sources. Unlike other methods, FastST exhibited good scalability, meaning that it effectively maintained consistent run-time and accuracy regardless of increasing complexity in terms of data size and dimensionality.

One of the long-standing obstacles to the practical use of microbial source tracking lies in computational efficiency. As shown in the simulation study, when the complexity of microbial communities increases, MST methods usually encounter significant computational difficulties. Due to its Bayesian setup and non-negative matrix factorization, SourceTracker2 and SourceID-NMF are computationally expensive. FEAST and STENSL, while significantly faster than SourceTracker2 at a lower number of known sources, suffered a drastic increase in computation time with a larger dataset. FastST, on the other hand, maintained stable performance with minimal runtime increase, demonstrating suitability for practical applications involving a large number of sources. Additionally, FastST’s capability to accurately infer directionality without predefined source–sink relationships enhances its applicability in real-world scenarios, where distinguishing sources from sinks can be challenging. The high accuracy rates achieved in identifying the correct source–sink relationship underscore its potential for various ecological and public health applications.

It is noteworthy that, our simulation study sets the number of taxa N to be greater than the number of source samples K, which is intuitively reasonable for real applications. When this assumption is not satisfied, for example, there exists a large number of source samples, a potential solution is to conduct a dimension reduction (e.g., PCA) before building a mixed-proportion model for the sink-source relationships. After all, when K is large, both the parameter of the taxa relative abundances γij and the parameter of proportions αi become difficult to estimate. As a rule of thumb, when K is greater than 100, most of the minor sources have very tiny, negligible contributions to the source, making the inference on αi worthless.

In conclusion, FastST has a potential to offer a substantial advancement in microbial source tracking by providing rapid and precise estimations and reliably inferring directionality. However, different scenario cases in simulation testing reveals that FastST has robust performance in both accuracy and computational efficiency, and also provide a directionality inference method which hasn’t been firstly presented to the best of our knowledge. Future research should explore extending FastST to incorporate more complex environmental scenarios with real datasets.

Supplemental Information

Supplemental Information 1 Average Jensen-Shannon divergence and Pearson correlation for estimating the proportion of sources in fully simulated microbiome data across MST methods

Supplemental Information 2 Average Jensen-Shannon divergence and Pearson correlation for estimating the proportion of sources in semi-synthetic microbiome data across MST methods

Supplemental Information 3 Tutorial for the FastST software

We acknowledge the use of ChatGPT to improve the grammar and the writing style of the manuscript.

Additional Information and Declarations

Competing Interests

Author Contributions

Data Availability

The authors declare there are no competing interests.

Joung Min Choi conceived and designed the experiments, performed the experiments, analyzed the data, prepared figures and/or tables, authored or reviewed drafts of the article, and approved the final draft.

Xiaowei Wu conceived and designed the experiments, performed the experiments, analyzed the data, prepared figures and/or tables, authored or reviewed drafts of the article, and approved the final draft.

Liqing Zhang conceived and designed the experiments, authored or reviewed drafts of the article, and approved the final draft.

The following information was supplied regarding data availability:

The data and code are available at GitHub and Zenodo:

- https://github.com/joungmin-choi/FastST

- Joung Min Choi. (2025). joungmin-choi/FastST: FastST v1.1.0 (v1.1.0). Zenodo. https://doi.org/10.5281/zenodo.16880909

The Knights et al. dataset is available at: https://github.com/danknights/sourcetracker (DOI: 10.1038/nmeth.1650).

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
