# Peer review of "FastST: an efficient tool for inferring decomposition and directionality of microbial communities"

_PeerJ, doi:10.7717/peerj.20161_

## Round 0.1 · original submission · Major Revisions

Your manuscript has been reviewed by two external experts. While they found your work interesting, they also raised significant concerns regarding the simulation runs, reproducibility, and the inference accuracy of the tool. Please address these issues, along with the other comments provided by the reviewers, in your revised submission.

Reviewer 1 ·

Basic reporting

This paper seems very promising and makes a lot of claims about being a huge speed and computational improvement, but this was not the case for me.
I downloaded the raw data files directly using R version 4.2.2 npreg_1.1.0 gtools_3.9.5 which are all acceptable according to the instructions) and tried the simulations.
R version 4.2.2 (2022-10-31)
Platform: x86_64-apple-darwin17.0 (64-bit)
Running under: macOS 15.4.1
other attached packages:
[1] npreg_1.1.0 gtools_3.9.5
I excuted simu1.R on my laptop with 1000 simulations, but it ran all night and it never finished. I reduced it to 10 simulations and it ran for well over an hour and didn’t finish.
Separately, I tried to get it to run individual files to that I could test it with real-world example from the literature, but this program was not designed to read files which makes it less than useful for others. Given the great difficulty I had even running a few simulations using the code straight from github and the authors instructions, I don’t believe this software is ready for prime time. It is also missing a lot of nice output of the other programs.

Experimental design

I wasn't able to run the simulations following the instructions to the letter and I'm not sure about the parameters used for software comparisons. The lack of comparisons on real-world data (outside of simulations) is also problematic for me, esp. since the authors say this approach is better in the "real-world".

Validity of the findings

Unclear since I coudn't run the simulations or easily test alternative datasets with know answers.

Additional comments

Other issues:
1) The authors make a big deal about how SourceTracer is great for “real-world” situations where we don’t know which is the source and which is the sink (or the direction). But I’ve been doing studies for some time and most of the time we do know the source. I’ve looked at well over 100 papers doing sourcetracker work and the sinks are easy to identify in most cases. The authors should cite some studies where this is not the case - where it is a real question which is the sink. I can think of some situations where one might want to know how much travels *between* environment but I want to see some examples where this is completely unknown.
2) There is no information on how the authors did the direct comparisons with SourceTracker and FEAST – they don’t make it easy to reproduce these results. It is one thing to show graphs, but what were the parameters used in ST (for example)? ST2 is also available – see the MetaSourceTracker paper, and it can be parallelized. But also, it can be run using multiple Markov chains and longer chains (restarts) which reduces the error estimates considerable. It looks like the authors used the R code from 2011, but the authors should use the new version and show how they used it for a proper comparison. The code for running test files and the parameters for ST and FEAST should be included.
3) For the SourceTracer, I would at least expect some of the basic outputs that ST presents in terms of source proportion graphs (pdfs) with error bars. The software should be useful for other datasets outside of simulations and produce readable output or output easy to put into other programs.
4) In terms of the code, outside of the simulations it is not clear how to use SourceTracer with other datasets. How are researchers supposed to use it instead of the original ST or FEAST? There should be an easy way to input original datasets from studies.
5) The name SourceTracer is so similar to SourceTracker that they could easily be confused. They differ by only one letter which could lead to conflation but also spelling issues. Maybe a name like SourceIdentifier or SourceDetector might be better. It shouldn’t be too hard to make a new repository with a different name.

Reviewer 2 ·

Basic reporting

This paper introduces a new microbial source tracking tool, SourceTracer. The authors clearly identify two key contributions of this tool: its computational efficiency and its ability to infer the direction between sources. The simulated experiments demonstrate SourceTracer's performance in these areas to some extent, particularly regarding its computational efficiency. However, I have concerns about inference accuracy, as many details are lacking, and the inference of direction between sources is not clearly articulated. Overall, I have the following comments:

Major:
1. One of the significant contributions of the tool is the inference of direction between sources. To my understanding, the tool determines the direction by maximizing the probabilities of K models that treat each source as the sink. However, the description is too concise to be fully understood. The authors should clearly explain why maximizing the probability can determine the direction and how this can be inferred from the estimated contributions.

2. Since the direction between sources is inferred by treating each source as a sink, it seems that FEAST and SourceTracker could adopt a similar approach. By considering each sample as a sink and the remaining samples as potential known sources, these tools may determine the direction between sources by analyzing the contribution values. For example, if the estimated contribution of source A to sink B is non-empty, while the contribution of sink B to source A is either empty or significantly smaller, we can infer a directional relationship from A to B. This is especially relevant when the actual sink sample contains more taxa than those contributed by source A, indicating that some taxa may not exist in source A at all. In such cases, the unique taxa present in the sink from other sources can help clarify the direction of contribution, allowing us to validate the estimations directly. While this approach may be time-consuming for FEAST and SourceTracker, it is worth considering, especially when there are fewer sources available for inference. I would recommend that the authors compare FEAST and SourceTracker in terms of direction inference.

3. Many details about the simulated data are missing. Did you use the real microbial taxa distribution (e.g., from the Earth Microbiome Project) to simulate the data? How similar is the taxa distribution between sources, especially the major sources? When the sources have similar taxa distributions, will this tool still work well?

4. How do you simulate noise in the reference sources? The taxa distribution of the contributed sources may differ slightly from the observed taxa distribution. Since this model directly utilizes the observed taxa distribution without inferring the distribution of the contributed sources, will the performance be significantly affected by this noise? Additionally, if some low-abundance taxa are absent in the observed sources (with zero abundance) but present in the sink source, will the model still perform well?

5. Is there no real data experiment for validation? While there may not be real data with a known ground truth, comparing the similarities and differences in the results obtained from various tools using actual data can help validate the model's capabilities to some extent.

6. There are a couple of other tools that are missed by the authors, such as STENSL and SourceID-NMF. Please compare with them.

Minor:
1. MAE is a useful metric for evaluating the accuracy of estimated source contributions. However, it does not assess the accuracy of trend estimation. Since the differences in MAE among the tools are very minor, I recommend using the Pearson correlation of the Jensen-Shannon divergence to further evaluate performance, like the approach taken by FEAST.

Experimental design

please see box 1.

Validity of the findings

please see box 1.

Additional comments

please see box 1.

---

## Round 0.2 · accepted · Accept

Based on the opinion of an additional reviewer and my own assessment of the revision, I can confirm that the authors have addressed all of the reviewers' comments. I am satisfied with the current version and believe it is ready for publication.

Reviewer 3 ·

Basic reporting

The authors have reasonably addressed my earlier comments, and the revised manuscript has improved significantly.

Minors:
A typo at line 251 in page 8: 'ans'. Should be 'and'.

Experimental design

no comment

Validity of the findings

no comment

Additional comments

no comment